# CONTINGENCY-AWARE EXPLORATION IN REINFORCEMENT LEARNING

**Jongwook Choi**[*,1]    **Yijie Guo**[*,1]    **Marcin Moczulski**[*,2]
**Junhyuk Oh**[1,†]    **Neal Wu**[2]    **Mohammad Norouzi**[2]    **Honglak Lee**[2,1]
[1]University of Michigan        [2]Google Brain
{jwook,guoyijie}@umich.edu   moczulski@google.com
{junhyuk,nealwu,mnorouzi,honglak}@google.com

## ABSTRACT

This paper investigates whether learning contingency-awareness and controllable aspects of an environment can lead to better exploration in reinforcement learning. To investigate this question, we consider an instantiation of this hypothesis evaluated on the Arcade Learning Element (ALE). In this study, we develop an attentive dynamics model (ADM) that discovers controllable elements of the observations, which are often associated with the location of the character in Atari games. The ADM is trained in a self-supervised fashion to predict the actions taken by the agent. The learned contingency information is used as a part of the state representation for exploration purposes. We demonstrate that combining actor-critic algorithm with count-based exploration using our representation achieves impressive results on a set of notoriously challenging Atari games due to sparse rewards.[1] For example, we report a state-of-the-art score of >11,000 points on MONTEZUMA'S REVENGE without using expert demonstrations, explicit high-level information (*e.g.*, RAM states), or supervisory data. Our experiments confirm that contingency-awareness is indeed an extremely powerful concept for tackling exploration problems in reinforcement learning and opens up interesting research questions for further investigations.

## 1    INTRODUCTION

The success of reinforcement learning (RL) algorithms in complex environments hinges on the way they balance *exploration* and *exploitation*. There has been a surge of recent interest in developing effective exploration strategies for problems with high-dimensional state spaces and sparse rewards (Schmidhuber, 1991; Oudeyer & Kaplan, 2009; Houthooft et al., 2016; Bellemare et al., 2016; Osband et al., 2016; Pathak et al., 2017; Plappert et al., 2018; Zheng et al., 2018). Deep neural networks have seen great success as expressive function approximators within RL and as powerful representation learning methods for many domains. In addition, there have been recent studies on using neural network representations for exploration (Tang et al., 2017; Martin et al., 2017; Pathak et al., 2017). For example, count-based exploration with neural density estimation (Bellemare et al., 2016; Tang et al., 2017; Ostrovski et al., 2017) presents one of the state-of-the-art techniques on the most challenging Atari games with sparse rewards.

Despite the success of recent exploration methods, it is still an open question on how to construct an optimal representation for exploration. For example, the concept of visual similarity is used for learning density models as a basis for calculating *pseudo-counts* (Bellemare et al., 2016; Ostrovski et al., 2017). However, as Tang et al. (2017) noted, the ideal way to represent states should be based on what is relevant to solving the MDP, rather than only relying on visual similarity. In addition, there remains another question on whether the representations used for recent exploration works are easily interpretable. To address these questions, we investigate whether we can learn a complementary, more intuitive, and interpretable high-level abstraction that can be very effective in exploration by using the ideas of contingency awareness and controllable dynamics.

---

[*]Equal contributions, listed in alphabetical order.
[†]Now at DeepMind.
[1]Examples of the learned policy and the contingent regions are available at https://coex-rl.github.io/.

The key idea that we focus on in this work is the notion of *contingency awareness* (Watson, 1966; Bellemare et al., 2012) — the agent's understanding of the environmental dynamics and recognizing that some aspects of the dynamics are under the agent's control. Intuitively speaking, this can represent the segmentation mask of the agent operating in the 2D or 3D environments (yet one can think of more abstract and general state spaces). In this study, we investigate the concept of contingency awareness based on *self-localization*, *i.e.*, the awareness of where the agent is located in the abstract state space. We are interested in discovering parts of the world that are directly dependent on the agent's immediate action, which often reveal the agent's approximate location.

For further motivation on the problem, we note that contingency awareness is a very important concept in neuroscience and psychology. In other words, being self-aware of one's location is an important property within many observed intelligent organisms and systems. For example, recent breakthroughs in neuroscience, such as the Nobel Prize winning work on the grid cells (Moser et al., 2015; Banino et al., 2018), show that organisms that perform very well in spatially-challenging tasks are self-aware of their location. This allows rats to navigate, remember paths to previously visited places and important sub-goals, and find shortcuts. In addition, the notion of contingency awareness has been shown as an important factor in developmental psychology (Watson, 1966; Baeyens et al., 1990). We can think of self-localization (and more broadly self-awareness) as a principled and fundamental direction towards intelligent agents.

Based on these discussions, we hypothesize that contingency awareness can be a powerful mechanism for tackling exploration problems in reinforcement learning. We consider an instantiation of this hypothesis evaluated on the Arcade Learning Element (ALE). For example, in the context of 2D Atari games, contingency-awareness roughly corresponds to understanding the notion of controllable entities (*e.g.*, the player's avatar), which Bellemare et al. (2012) refer to as *contingent regions*. More concretely, as shown in Figure 1, in the game FREEWAY, only the chicken sprite is under the agent's control and not the multiple moving cars; therefore the chicken's location should be an informative element for exploration (Bellemare et al., 2012; Pathak et al., 2017).

In this study, we also investigate whether contingency awareness can be learned without any external annotations or supervision. For this, we provide an instantiation of an algorithm for automatically learning such information and using it for improving exploration on a 2D ALE environment (Bellemare et al., 2013). Concretely, we employ an *attentive dynamics model* (ADM) to predict the agent's action chosen between consecutive states. It allows us to approximate the agent's position in 2D environments, but unlike other approaches such as (Bellemare et al., 2012), it does not require any additional supervision to do so. The ADM learns in an online and self-supervised fashion with pure observations as the agent's policy is updated and does not require hand-crafted features, an environment simulator, or supervision labels for training.

In experimental evaluation, our methods significantly improve the performance of A2C on hard-exploration Atari games in comparison with competitive methods such as density-based exploration (Bellemare et al., 2016; Ostrovski et al., 2017) and SimHash (Tang et al., 2017). We report very strong results on sparse-reward Atari games, including the state-of-the-art performance on the notoriously difficult MONTEZUMA'S REVENGE, when combining our proposed exploration strategy with PPO (Schulman et al., 2017), without using expert demonstrations, explicit high-level information (*e.g.*, RAM states), or resetting the environment to an arbitrary state.

We summarize our contributions as follows:

- We demonstrate the importance of learning contingency awareness for efficient exploration in challenging sparse-reward RL problems.
- We develop a novel instance of attentive dynamics model using contingency and controllable dynamics to provide robust localization abilities across the most challenging Atari environments.
- We achieve a strong performance on difficult sparse-reward Atari games, including the state-of-the-art score on the notoriously challenging MONTEZUMA'S REVENGE.

Overall, we believe that our experiments confirm the hypothesis that contingency awareness is an extremely powerful concept for tackling exploration problems in reinforcement learning, which opens up interesting research questions for further investigations.

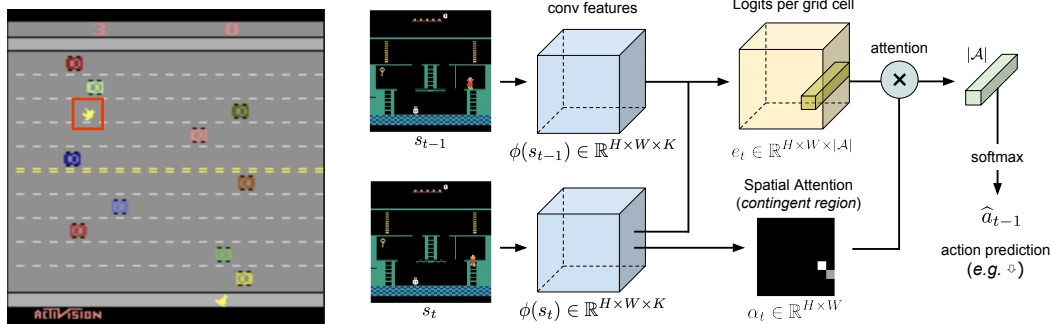

Figure 1: **Left:** Contingent region in FREEWAY; an object in a red box denotes what is under the agent's control, whereas the rest is not. **Right:** A diagram for the proposed ADM architecture.

## 2 RELATED WORK

**Self-Localization.** The discovery of grid cells (Moser et al., 2015) motivates working on agents that are self-aware of their location. Banino et al. (2018) emphasize the importance of self-localization and train a neural network which learns a similar mechanism to grid cells to perform tasks related to spatial navigation. The presence of grid cells is correlated with high performance. Although grid cells seem tailored to 2D or 3D problems that animals encounter in their life, it is speculated that their use can be extended to more abstract spaces. A set of potential approaches to self-localization ranges from ideas specific to a given environment, *e.g.*, SLAM (Durrant-Whyte & Bailey, 2006), to methods with potential generalizability (Mirowski et al., 2017; Jaderberg et al., 2017; Mirowski et al., 2018).

**Self-supervised Dynamics Model and Controllable Dynamics.** Several works have used forward and/or inverse dynamics models of the environment (Oh et al., 2015; Agrawal et al., 2016; Shelhamer et al., 2017). Pathak et al. (2017) employ a similar dynamics model to learn feature representations of states that captures controllable aspects of the environment. This dense representation is used to design a curiosity-driven intrinsic reward. The idea of learning representations on relevant aspects of the environment by learning auxiliary tasks is also explored in (Jaderberg et al., 2017; Bengio et al., 2017; Sawada, 2018). Our presented approach is different as we focus on explicitly discovering controllable aspects using an attention mechanism, resulting in better interpretability.

**Exploration and Intrinsic Motivation.** The idea of providing an exploration bonus reward depending on the state-action visit-count was proposed by Strehl & Littman (2008) (MBIE-EB), originally under a tabular setting. Later it has been combined with different techniques to deal with high-dimensional state spaces. Bellemare et al. (2016) use a Context-Tree Switching (CTS) density model to derive a state *pseudo-count*, whereas Ostrovski et al. (2017) use PixelCNN as a state density estimator. Martin et al. (2017) also construct a visitation density model over a compressed feature space rather than the raw observation space. Alternatively, Tang et al. (2017) propose a locality-sensitive hashing (LSH) method to cluster states and maintain a state-visitation counter based on a form of similarity between frames. We train an agent with a similar count-based exploration bonus, but the way of maintaining state counter seems relatively simpler in that key feature information (*i.e.*, controllable region) is explicitly extracted from the observation and directly used for counting states.

Another popular family of exploration strategies in RL uses intrinsic motivation (Schmidhuber, 1991; Singh et al., 2004; Oudeyer & Kaplan, 2009; Barto, 2013). These methods encourage the agent to look for something surprising in the environment which motivates its search for novel states, such as surprise (Achiam & Sastry, 2017), curiosity (Pathak et al., 2017; Burda et al., 2018), and diversity (Eysenbach et al., 2018), or via feature control (Jaderberg et al., 2017; Dilokthanakul et al., 2017).

## 3 APPROACH

### 3.1 DISCOVERING CONTINGENCY VIA ATTENTIVE DYNAMICS MODEL

To discover the region of the observation that is controllable by the agent, we develop an instance of *attentive dynamics model* (ADM) based on inverse dynamics $f_{\text{inv}}$. The model takes two consecutive input frames (observations) $s_{t-1}, s_t \in \mathcal{S}$ as input and aims to predict the action ($a_{t-1} \in \mathcal{A}$) taken by

the agent to transition from $s_{t-1}$ to $s_t$:

$$\widehat{a}_{t-1} = f_{\text{inv}}(s_{t-1}, s_t). \tag{1}$$

Our key intuition is that the inverse dynamics model should attend to the most relevant part of the observation, which is controllable by the agent, to be able to classify the actions. We determine whether each region in a $H \times W$ grid is controllable, or in other words, useful for predicting the agent's action, by using a spatial *attention mechanism* (Bahdanau et al., 2015; Xu et al., 2015). An overview of the model is shown in Figure 1.

**Model.** To perform action classification, we first compute a convolutional feature map $\phi_t^s = \phi(s_t) \in \mathbb{R}^{H \times W \times K}$ based on the observation $s_t$ using a convolutional neural network $\phi$. We estimate a set of *logit* (score) vectors, denoted $e_t(i,j) \in \mathbb{R}^{|\mathcal{A}|}$, for action classification from each grid cell $(i,j)$ of the convolutional feature map. The local convolution features and feature differences for consecutive frames are fed into a shared multi-layer perceptron (MLP) to derive the logits as:

$$e_t(i,j) = \text{MLP}\Big( \big[ \phi_t^s(i,j) - \phi_{t-1}^s(i,j); \ \phi_t^s(i,j) \big] \Big) \in \mathbb{R}^{|\mathcal{A}|}. \tag{2}$$

We then compute an attention mask $\alpha_t \in \mathbb{R}^{H \times W}$ corresponding to frame $t$, which indicates the controllable parts of the observation $s_t$. Such attention masks are computed via a separate MLP from the features of each region $(i,j)$, and then converted into a probability distribution using softmax or sparsemax operators (Martins & Astudillo, 2016):

$$\alpha_t = \text{sparsemax}(\widetilde{\alpha}_t) \quad \text{where} \quad \widetilde{\alpha}_t(i,j) = \text{MLP}\big(\phi_t^s(i,j)\big), \tag{3}$$

so that $\sum_{i,j} \alpha_t(i,j) = 1$. The sparsemax operator is similar to softmax but yields a sparse attention, leading to more stable performance. Finally, the logits $e_t(i,j)$ from all regions are linearly combined using the attention probabilities $\alpha_t$:

$$p(\widehat{a}_{t-1} \mid s_{t-1}, s_t) = \text{softmax}\Big( \sum_{i,j} \alpha_t(i,j) \cdot e_t(i,j) \Big) \in \mathbb{R}^{|\mathcal{A}|}. \tag{4}$$

**Training.** The model can be optimized with the standard cross-entropy loss $\mathcal{L}_{\text{action}}(a_{t-1}^*, \widehat{a}_{t-1})$ with respect to the ground-truth action $a_{t-1}^* \in \mathcal{A}$ that the agent actually has taken. Based on this formulation, the attention probability $\alpha_t(i,j)$ should be high only on regions $(i,j)$ that are predictive of the agent's actions. Our formulation enables learning to localize controllable entities in a self-supervised way without any additional supervisory signal, unlike some prior work (*e.g.*, (Bellemare et al., 2012)) that adopts simulators to collect extra supervisory labels.

Optimizing the parameters of ADM on on-policy data is challenging for several reasons. First, the ground-truth action may be unpredictable for given pairs of frames, leading to noisy labels. For example, actions taken in uncontrollable situations do not have any effect (*e.g.*, when the agent is in the middle of jumping in MONTEZUMA'S REVENGE). Second, since we train the ADM online along with the policy, the training examples are not independently and identically distributed, and the data distribution can shift dramatically over time. Third, the action distribution from the agent's policy can run into a low entropy[2], being biased towards certain actions. These issues may prevent the ADM from generalization to novel observations, which hurts exploration. Generally, we prefer models that quickly adapt to the policy and learn to localize the controllable regions in a robust manner.

To mitigate the aforementioned issues, we adopt a few additional objective functions. We encourage the attention distribution to attain a high entropy by including an *attention entropy regularization loss*, *i.e.*, $\mathcal{L}_{\text{ent}} = -\mathcal{H}(\alpha_t)$. This term penalizes over-confident attention masks, making the attention closer to uniform whenever action prediction is not possible. We also train the logits corresponding to each grid cell independently using a separate cross-entropy loss: $p(\widehat{a}_{t-1}^{i,j} \mid e_t(i,j)) = \text{softmax}(e_t(i,j))$. These additional cross-entropy losses, denoted $\mathcal{L}_{\text{cell}}^{i,j}$, allow the model to learn from unseen observations even when attention fails to perform well at first. The entire training objective becomes:

$$\mathcal{L}^{\text{ADM}} = \mathcal{L}_{\text{action}} + \sum_{i,j} \mathcal{L}_{\text{cell}}^{i,j} + \lambda_{\text{ent}} \mathcal{L}_{\text{ent}} \tag{5}$$

where $\lambda_{\text{ent}}$ is a mixing hyperparameter.

## 3.2 COUNT-BASED EXPLORATION WITH CONTINGENT REGIONS

One natural way to take advantage of discovered contingent regions for exploration is count-based exploration. The ADM can be used to localize the controllable entity (*e.g.*, the agent's avatar)

---

[2]We note that an entropy regularization term (*e.g.*, Eq.(9)) is used when learning the policy.

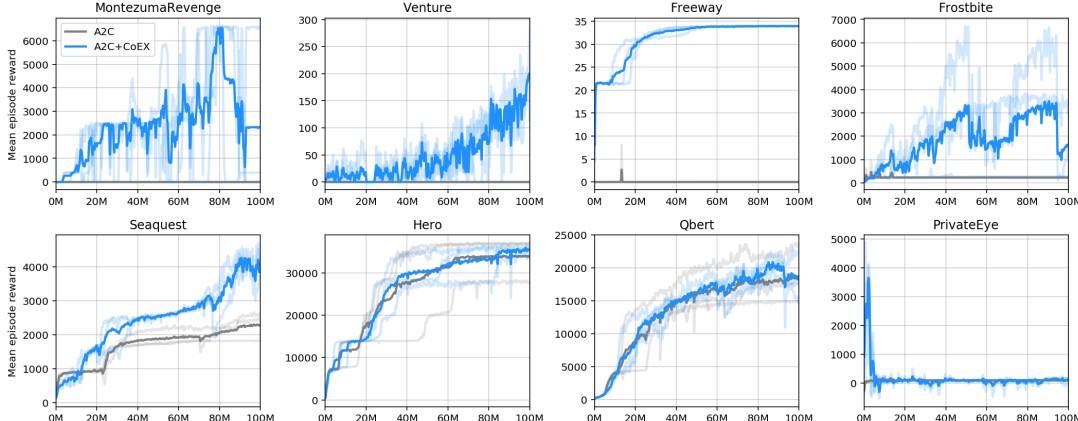

Figure 2: Learning curves on several Atari games: A2C+CoEX and A2C. The x-axis represents total environment steps and the y-axis the mean episode reward averaged over 40 recent episodes. The mean curve is obtained by averaging over 3 random seeds, each shown in a light color.

from an observation $s_t$ experienced by the agent. In 2D environments, a natural discretization $(x, y) = \mathrm{argmax}_{(j,i)} \, \alpha_t(i, j)$ provides a good approximation of the agent's location within the current observation[3]. This provides a key piece of information about the current state of the agent.

Inspired by previous work (Bellemare et al., 2016; Tang et al., 2017), we add an exploration bonus of $r^+$ to the environment reward, where $r^+(s) = 1/\sqrt{\#(\psi(s))}$ and $\#(\psi(s))$ denotes the visitation count of the (discrete) mapped state $\psi(s)$, which consists of the contingent region $(x, y)$. We want to find a policy $\pi$ that maximizes the expected discounted sum of environment rewards $r^{\mathrm{ext}}$ plus count-based exploration rewards $r^+$, denoted $\mathcal{R} = \mathbb{E}_\pi \big[ \sum_t \gamma^t \big( \beta_1 r^{\mathrm{ext}}(s_t, a_t) + \beta_2 r^+(s_t) \big) \big]$, where $\beta_1, \beta_2 \geq 0$ are hyperparameters that balance the weight of environment reward and exploration bonus. For every state $s_t$ encountered at time step $t$, we increase the counter value $\#(\psi(s_t))$ by 1 during training. The full procedure is summarized in Algorithm 1 in Appendix A.

## 4 EXPERIMENTS

In the experiments below we investigate the following key questions:

- Does the contingency awareness in terms of self-localization provide a useful state abstraction for exploration?
- How well can the self-supervised model discover the ground-truth abstract states?
- How well does the proposed exploration strategy perform against other exploration methods?

### 4.1 EXPERIMENTS WITH A2C

We evaluate the proposed exploration strategy on several difficult exploration Atari 2600 games from the Arcade Learning Environment (ALE) (Bellemare et al., 2013). We focus on 8 Atari games including FREEWAY, FROSTBITE, HERO, PRIVATEEYE, MONTEZUMA'S REVENGE, QBERT, SEAQUEST, and VENTURE. In these games, an agent without an effective exploration strategy can often converge to a suboptimal policy. For example, as depicted in Figure 2, the Advantage Actor-Critic (A2C) baseline (Mnih et al., 2016) achieves a reward close to 0 on MONTEZUMA'S REVENGE, VENTURE, FREEWAY, FROSTBITE, and PRIVATEEYE, even after 100M steps of training. By contrast, our proposed technique, which augments A2C with count-based exploration with the location information learned by the attentive dynamics model, denoted **A2C+CoEX** (CoEX stands for "Contingency-aware Exploration"), significantly outperforms the A2C baseline on six out of the 8 games.

We compare our proposed A2C+CoEX technique against the following baselines:[4]

---

[3]To obtain more accurate localization by taking temporal correlation into account, we can use exponential smoothing as $\overline{\alpha}_t(i, j) = (1 - \omega_t)\overline{\alpha}_{t-1}(i, j) + \omega_t \alpha_t(i, j)$, where $\omega_t = \max_{(i,j)}\{\alpha_t(i, j)\}$.

[4]In Section 4.6, we also report experiments using Proximal Policy Optimization (PPO) (Schulman et al., 2017) as a baseline, where our **PPO+CoEX** achieves the average score of >11,000 on MONTEZUMA'S REVENGE.

| Method | Freeway | Frostbite | Hero | Montezuma | PrivateEye | Qbert | Seaquest | Venture |
|---|---|---|---|---|---|---|---|---|
| A2C | 7.2 | 1099 | 34352 | 13 | 574 | 19620 | 2401 | 0 |
| A2C+Pixel-SimHash | 0.0 | 829 | 28181 | 412 | 276 | 18180 | 2177 | 31 |
| A2C+CoEX | **34.0** | **4260** | **36827** | **6635** | **5316** | **23962** | **5169** | **204** |
| A2C+CoEX+RAM* | 34.0 | 4418 | 36765 | 6600 | 24296 | 24422 | 6113 | 1100 |

Table 1: Performance of our method and its baselines on Atari games: maximum mean scores (averaged over 40 recent episodes) achieved over total 100M environment timesteps (400M frames) of training, averaged over 3 seeds. The best entry in the group of experiments without supervision is shown in bold. * denotes that A2C+CoEX+RAM acts as a control experiment, which includes some supervision. More experimental results on A2C+CoEX+RAM are shown in Appendix C.

| Method | #Steps | Freeway | Frostbite | Hero | Montezuma | PrivateEye | Qbert | Seaquest | Venture |
|---|---|---|---|---|---|---|---|---|---|
| A2C+CoEX (Ours) | 50M | 33.9 | 3900 | 31367 | 4100 | 5316 | 17724 | 2620 | 128 |
| A2C+CoEX (Ours) | 100M | **34.0** | 4260 | **36827** | **6635** | 5316 | **23962** | **5169** | 204 |
| DDQN+ | 25M | 29.2 | - | 20300 | 3439 | 1880 | - | - | 369 |
| A3C+ | 50M | 27.3 | 507 | 15210 | 142 | 100 | 15805 | 2274 | 0 |
| TRPO-AE-SimHash | 50M | 33.5 | **5214** | - | 75 | - | - | - | 445 |
| Sarsa-$\phi$-EB | 25M | 0.0 | 2770 | - | 2745 | - | 4112 | - | 1169 |
| DQN-PixelCNN | 37.5M | 31.7 | - | - | 2514 | **15806** | 5501 | - | **1356** |
| Curiosity-Driven | 25M | 32.8 | - | - | 2505 | 3037 | - | - | 416 |

Table 2: Performance of our method and state-of-the-art exploration methods on Atari games. For fair comparison, we report the maximum mean score achieved over the specific number of timesteps during training, averaged over 3 seeds. The best entry is shown in bold. Baselines (for reference) are: DDQN+ and A3C+ (Bellemare et al., 2016), TRPO-AE-SimHash (Tang et al., 2017), Sarsa-$\phi$-EB (Martin et al., 2017), DQN-PixelCNN (Ostrovski et al., 2017), and Curiosity-Driven (Burda et al., 2018). The numbers for DDQN+ were taken from (Tang et al., 2017) or were read from a plot.

- **A2C**: an implementation adopted from OpenAI baselines (Dhariwal et al., 2017) using the default hyperparameters, which serves as the building block of our more complicated baselines.
- **A2C+Pixel-SimHash**: Following (Tang et al., 2017), we map 52×52 gray-scale observations to 128-bit binary codes using random projection followed by quantization (Charikar, 2002). Then, we add a count-based exploration bonus based on quantized observations.

As a control experiment, we evaluate **A2C+CoEX+RAM***, our contingency-aware exploration method together with the ground-truth location information obtained from game's RAM. It is roughly an upper-bound of the performance of our approach.

## 4.2 Implementation Details

For the A2C (Mnih et al., 2016) algorithm, we use 16 parallel actors to collect the agent's experience, with 5-step rollout, which yields a minibatch of size 80 for on-policy transitions. We use the last 4 observation frames stacked as input, each of which is resized to $84 \times 84$ and converted to grayscale as in (Mnih et al., 2015; 2016). We set the end of an episode to when the game ends, rather than when the agent loses a life. Each episode is initialized with a random number of no-ops (Mnih et al., 2015). More implementation details can be found in Appendix A and B.

For the ADM, we take observation frames of size $160 \times 160$ as input (resized from the raw observation of size $210 \times 160$).[5] We employ a 4-layer convolutional neural network that produces a feature map $\phi(s_t)$ with a spatial grid size of $H \times W = 9 \times 9$. As a result, the prediction of location coordinates lies in the $9 \times 9$ grid.

In some environments, the contingent regions within the visual observation alone are not sufficient to determine the exact location of the agent within the game; for example, the coordinate cannot

---

[5] In some games such as Venture, the agent is depicted in very small pixels, which might be hardly recognizable in rescaled $84 \times 84$ images.

Figure 3: Performance plot of ADM trained using on-policy samples from the A2C+CoEX agent.

solely distinguish between different rooms in HERO, MONTEZUMA'S REVENGE, and PRIVATEEYE, etc. Therefore, we introduce a discrete context representation $c \in \mathbb{Z}$ that summarizes the high-level visual context in which the agent currently lies. We use a simple clustering method similar to (Kulis & Jordan, 2012), which we refer to as *observation embedding clustering* that clusters the random projection vectors of the input frames as in (Tang et al., 2017), so that different contexts are assigned to different clusters. We further explain this heuristic approach more in detail in Appendix D.

In sparse-reward problems, the act of collecting a reward is rare but frequently instrumental for the future states of the environment. The cumulative reward $R_t = \sum_{t'=0}^{t-1} r^{\text{ext}}(s_{t'}, a_{t'})$ from the beginning of the episode up to the current step $t$, can provide a useful high-level *behavioral context* because collecting rewards can trigger significant changes to the agent's state and as a result the optimal behavior can change as well. In this sense, the agent should revisit the previously visited location for exploration when the context changes. For example, in MONTEZUMA'S REVENGE, if the agent is in the first room and the cumulative reward is 0, we know the agent has not picked up the key and the optimal policy is to reach the key. However, if the cumulative reward in the first room is 100, it means the agent has picked up the key and the next optimal goal is to open a door and move on to the next room. Therefore, we could include the cumulative reward as a part of state abstraction for exploration, which leads to empirically better performance.

To sum up, for the purpose of count-based exploration, we utilize the location $(x, y)$ of the controllable entity (*i.e.*, the agent) in the current observation discovered by ADM (Section 3.1), a context representation $c \in \mathbb{Z}$ that denotes the high level visual context, and a cumulative environment reward $R \in \mathbb{Z}$ that represents the exploration behavioral state. In such setting, we may denote $\psi(s) = (x, y, c, R)$.

### 4.3 PERFORMANCE OF COUNT-BASED EXPLORATION

Figure 2 shows the learning curves of the proposed methods on 8 Atari games. The performance of our method A2C+CoEX and A2C+CoEX+RAM as well as the baselines A2C and A2C+Pixel-SimHash are summarized in Table 1. In order to find a balance between the environment reward and the exploration bonus reward, we perform a hyper-parameter search for the proper weight of the environment reward $\beta_1$ and the exploration reward $\beta_2$ for A2C+CoEX+RAM, as well as for A2C+CoEX. The hyper-parameters for the two ended up being the same, which is consistent with our results. For fair comparison, we also search for the proper weight of environment reward for A2C baseline. The best hyper-parameters for each game are shown in Table 5 in Appendix B.

Compared to the vanilla A2C, the proposed exploration strategy improves the score on all the hard-exploration games. As shown in Table 1, provided the representation $(x, y, c, R)$ is perfect, A2C+CoEX+RAM achieves a significant improvement over A2C by encouraging the agent to visit novel locations, and could nearly solve these hard exploration games as training goes on.

Furthermore, A2C+CoEX using representations learned with our proposed attentive dynamics model and observation embedding clustering also outperforms the A2C baseline. Especially on FREEWAY, FROSTBITE, HERO, MONTEZUMA'S REVENGE, QBERT and SEAQUEST, the performance is comparable with A2C+CoEX+RAM, demonstrating the usefulness of the contigency-awareness information discovered by ADM.

**Comparison to other count-based exploration methods**. Table 2 compares the proposed method with previous state-of-the-art results, where our proposed method outperforms the other methods on 5 out of 8 games. DQN-PixelCNN is the strongest alternative achieving a state-of-the-art performance on some of the most difficult sparse-reward games. We argue that using Q-learning as the base learner with DQN-PixelCNN makes the direct comparison with A2C+CoEX not completely adequate. Note that the closest alternative count-based exploration method to A2C+CoEX would be A3C+ (Bellemare

Figure 4: Curves of ARI score during training of A2C+CoEX, averaged over 100 recent observations.

et al., 2016), which augments A3C (Mnih et al., 2016) with exploration bonus derived from pseudo-count, because A2C and A3C share a similar policy learning method. With that in mind, one can observe a clear improvement of A2C+CoEX over A3C+ on all of the 8 Atari games.

## 4.4 ANALYSIS OF ATTENTIVE DYNAMICS MODEL

We also analyze the performance of the ADM that learns the controllable dynamics of the environment. As a performance metric, we report the average distance between the ground-truth agent location $(x^*, y^*)$ and the predicted location $(x, y)$ within the $9 \times 9$ grid: $\|(x, y) - (x^*, y^*)\|_2$. The ground-truth location of the agent is extracted from RAM[6], then rescaled so that the observation image frame fits into the $9 \times 9$ grid.

Figure 3 shows the results on 4 Atari games (MONTEZUMA'S REVENGE, SEAQUEST, HERO, and VENTURE). The ADM is able to quickly capture the location of the agent without any supervision of localization, despite the agent constantly visiting new places. Typically the predicted location is on average 1 or 2 grid cells away from the ground-truth location. Whenever a novel scene is encountered (*e.g.*, the second room in MONTEZUMA'S REVENGE at around 10M steps), the average distance temporarily increases but quickly drops again as the model learns the new room. We provide videos of the agents playing and localization information as the supplementary material.[7]

## 4.5 ANALYSIS OF OBSERVATION EMBEDDING CLUSTERING

To make the agent aware of a change in high-level visual context (*i.e.*, rooms in Atari games) in some games such as MONTEZUMA'S REVENGE, VENTURE, HERO, and PRIVATEEYE, we obtain a representation of the high-level context and use it for exploration. The high-level visual contexts are different from each other (different layouts, objects, colors, etc.), so the embedding generated by a random projection is quite distinguishable and the clustering is accurate and robust.

For evaluation, given an observation in Atari games, we compare the discrete representation (*i.e.*, which cluster it is assigned to) based on the embedding from random projection to the ground-truth room number extracted from RAM. The Adjusted Rand Index (ARI) (Rand, 1971) measures the similarity between these two data clusterings. The ARI may only yield a value between 0 and 1, and is exactly 1 when the clusterings are identical.

The curves of the Adjusted Rand Index are shown in Figure 4. For MONTEZUMA'S REVENGE and VENTURE, the discrete representation as room number is roughly as good as the ground-truth. For HERO and PRIVATEEYE, since there are many rooms quite similar to one another, it is more challenging to accurately cluster the embeddings. The samples shown in Figure 7 in Appendix D show reasonable performances of the clustering method on all these games.

## 4.6 ADDITIONAL EXPERIMENTS WITH PPO

We also evaluate the proposed exploration algorithm on MONTEZUMA'S REVENGE using the sticky actions environment setup (Machado et al., 2017) identical to the setup found in (Burda et al., 2019). In the sticky action setup, the agent randomly repeats the previous action with probability of 0.25, preventing the algorithm from simply memorizing the correct sequence of actions and relying on determinism. The agent is trained with Proximal Policy Optimization (PPO) (Schulman et al., 2017) in conjunction with the proposed exploration method using 128 parallel actors to collect the experience. We used reward normalization and advantage normalization as in (Burda et al., 2018).

---

[6]Please note that the location from RAM is used only for analysis and evaluation purposes.
[7]A demo video of the learnt policy and localization is available at https://coex-rl.github.io/.

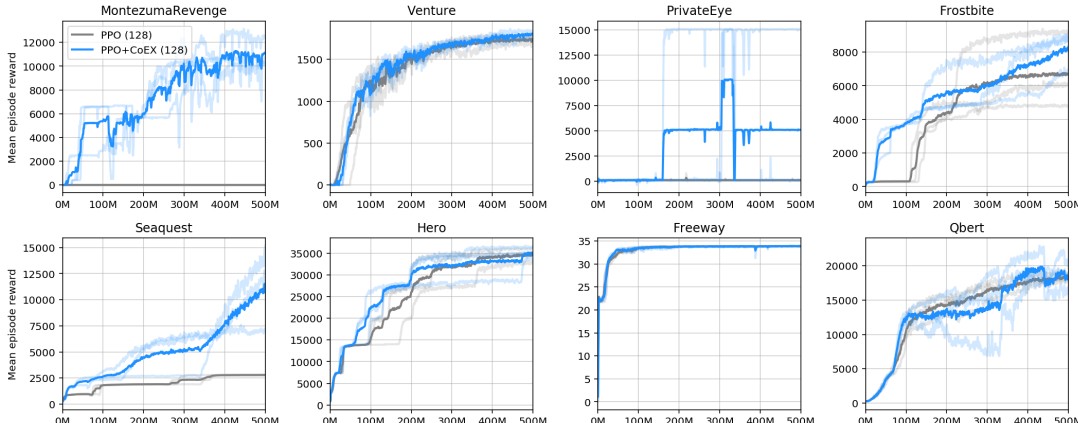

Figure 5: The learning curve of PPO+CoEX on several Atari games with sticky actions setup. The x-axis represents the total number of environment steps and the y-axis the mean episode reward averaged over 40 recent episodes. The mean curve is obtained by averaging over 3 random seeds, each shown in a light color.

| Method | #Steps | Freeway | Frostbite | Hero | Montezuma | PrivateEye | Qbert | Seaquest | Venture |
|--------|--------|---------|-----------|------|-----------|------------|-------|----------|---------|
| PPO | 500M | **34.0** | 7340 | 36263 | 29 | 942 | 19980 | 2806 | 1875 |
| PPO+CoEX | 500M | **34.0** | **9076** | **36664** | **11618** | **11000** | **22647** | **11794** | **1916** |

Table 3: Performance of PPO and PPO+CoEX: maximum mean scores (average over 40 recent episodes) achieved over total 500M environment steps (2B frames) of training, averaged over 3 seeds.

The method, denoted **PPO+CoEX**, achieves the score of 11,618 at 500M environment steps (2 billion frames) on MONTEZUMA'S REVENGE, when averaged over 3 runs. The learning curve is illustrated in Figure 5. Since the vanilla PPO baseline achieves a score near 0 (our runs) or 1,797 (Burda et al., 2019), this result is not solely due to the benefits of PPO. There is another approach "Exploration by Random Network Distillation" (Burda et al., 2019) concurrent with our work which achieves similar performance by following a slightly different philosophy.

## 4.7 DISCUSSIONS AND FUTURE WORK

This paper investigates whether discovering controllable dynamics via an attentive dynamics model (ADM) can help exploration in challenging sparse-reward environments. We demonstrate the effectiveness of this approach by achieving significant improvements on notoriously difficult video games. That being said, we acknowledge that our approach has certain limitations. Our currently presented instance of state abstraction method mainly focuses on controllable dynamics and employs a simple clustering scheme to abstract away uncontrollable elements of the scene. In more general setting, one can imagine using attentive (forward or inverse) dynamics models to learn an effective and compact abstraction of the controllable and uncontrollable dynamics as well, but we leave this to future work.

Key elements of the current ADM method include the use of spatial attention and modelling of the dynamics. These ideas can be generalized by a set of attention-based dynamics models (ADM) operating in forward, inverse, or combined mode. Such models could use attention over a lower-dimensional embedding that corresponds to an intrinsic manifold structure from the environment (*i.e.*, intuitively speaking, this also corresponds to *being self-aware of (e.g., locating) where the agent is in the abstract state space*). Our experiments with the inverse dynamics model suggest that the mechanism does not have to be perfectly precise, allowing for some error in practice. We speculate that mapping to such subspace could be obtained by techniques of embedding learning.

We note that RL environments with different visual characteristics may require different forms of attention-based techniques and properties of the model (*e.g.*, partial observability). Even though this paper focuses on 2D video games, we believe that the presented high-level ideas of learning contingency-awareness (with attention and dynamics models) are more general and could be applicable to more complex 3D environments with some extension. We leave this as future work.

## 5    CONCLUSION

We proposed a method of providing contingency-awareness through an attentive dynamics model (ADM). It enables approximate self-localization for an RL agent in 2D environments (as a specific perspective). The agent is able to estimate its position in the space and therefore benefits from a compact and informative representation of the world. This idea combined with a variant of count-based exploration achieves strong results in various sparse-reward Atari games. Furthermore, we report state-of-the-art results of >11,000 points on the infamously challenging MONTEZUMA'S REVENGE without using expert demonstrations or supervision. Though in this work we focus mostly on 2D environments in the form of sparse-reward Atari games, we view our presented high-level concept and approach as a stepping stone towards more universal algorithms capable of similar abilities in various RL environments.

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

# APPENDIX

## A SUMMARY OF TRAINING ALGORITHM

---

**Algorithm 1** A2C+CoEX

---

Initialize parameter $\theta_{\text{ADM}}$ for attentive dynamics model $f_{\text{ADM}}$
Initialize parameter $\theta_{\text{A2C}}$ for actor-critic network
Initialize parameter $\theta_c$ for context embedding projector if applicable (which is not trainable)
Initialize transition buffer $\mathcal{E} \leftarrow \emptyset$
**for** each iteration **do**
   ▷ *Collect on-policy transition samples, distributed over $K$ parallel actors*
   **for** each step $t$ **do**
      $s_t \leftarrow$ Observe state
      $a_t \sim \pi_\theta(a_t|s_t)$
      $s_{t+1}, r_t^{\text{ext}} \leftarrow$ Perform action $a_t$ in the environment
      ▷ *Compute the contingent region information*
      $\overline{\alpha}_{t+1} \leftarrow$ Compute the attention map of $s_{t+1}$ using $f_{\text{ADM}}$
      $c(s_{t+1}) \leftarrow$ Compute the observation embedding cluster of $s_{t+1}$ (Algorithm 2)
      ▷ *Increment state visitation counter based on the representation*
      $\psi(s_{t+1}) \leftarrow (\text{argmax}_{(i,j)} \, \overline{\alpha}_{t+1}(i,j), c(s_{t+1}), \lfloor \sum_{k=0}^{t} r_k^{\text{ext}} \rfloor)$
      $\#(\psi(s_{t+1})) \leftarrow \#(\psi(s_{t+1})) + 1$
      $r_t^+ \leftarrow \frac{1}{\sqrt{\#(\psi(s_{t+1}))}}$
      Store transition $\mathcal{E} \leftarrow \mathcal{E} \cup \left\{ (s_t, a_t, s_{t+1}, \beta_1 \text{clip}(r_t^{\text{ext}}, -1, 1) + \beta_2 r_t^+) \right\}$
   **end for**
   ▷ *Perform actor-critic using on-policy samples in $\mathcal{E}$*
   $\theta_{\text{A2C}} \leftarrow \theta_{\text{A2C}} - \eta \nabla_{\theta_{\text{A2C}}} \mathcal{L}^{\text{A2C}}$
   ▷ *Train the attentive dynamics model using on-policy samples in $\mathcal{E}$*
   $\theta_{\text{ADM}} \leftarrow \theta_{\text{ADM}} - \eta \nabla_{\theta_{\text{ADM}}} \mathcal{L}^{\text{ADM}}$
   Clear transition buffer $\mathcal{E} \leftarrow \emptyset$
**end for**

---

The learning objective $\mathcal{L}^{\text{ADM}}$ is from Equation (5). The objective $\mathcal{L}^{\text{A2C}}$ of Advantage Actor-Critic (A2C) is as in (Mnih et al., 2016; Dhariwal et al., 2017):

$$\mathcal{L}^{\text{A2C}} = \mathbb{E}_{(s,a,r) \sim \mathcal{E}} \left[ \mathcal{L}^{\text{A2C}}_{\text{policy}} + \frac{1}{2} \mathcal{L}^{\text{A2C}}_{\text{value}} \right] \tag{6}$$

$$\mathcal{L}^{\text{A2C}}_{\text{policy}} = -\log \pi_\theta(a_t|s_t)(R_t^n - V_\theta(s_t)) - \alpha \mathcal{H}_t(\pi_\theta) \tag{7}$$

$$\mathcal{L}^{\text{A2C}}_{\text{value}} = \frac{1}{2} \left( V_\theta(s_t) - R_t^n \right)^2 \tag{8}$$

$$\mathcal{H}_t(\pi_\theta) = -\sum_a \pi_\theta(a|s_t) \log \pi_\theta(a|s_t) \tag{9}$$

where $R_t^n = \sum_{i=0}^{n-1} \gamma^i r_{t+i} + \gamma^n V_\theta(s_{t+n})$ is the $n$-step bootstrapped return and $\alpha$ is a weight for the standard entropy regularization loss term $\mathcal{H}_t(\pi_\theta)$. We omit the subscript as $\theta = \theta_{\text{A2C}}$ when it is clear.

## B ARCHITECTURE AND HYPERPARAMETER DETAILS

The architecture details of the attentive dynamics model (ADM), the policy network, and hyper-parameters are as follows.

Table 4: Network architecture and hyperparameters

| | Hyperparameters | Value | |
|---|---|---|---|
| | Policy and Value Network Architecture | Input: 84x84x1 | |
| | | - Conv(32-8x8-4) | /ReLU |
| | | - Conv(64-4x4-2) | /ReLU |
| | | - Conv(64-3x3-1) | /ReLU |
| | | - FC(512) | /ReLU |
| | | - FC($|\mathcal{A}|$), FC(1) | |
| | ADM Encoder Architecture | Input: 160x160x3 | |
| | | - Conv(8-4x4-2) | /LeakyReLU |
| | | - Conv(8-3x3-2) | /LeakyReLU |
| | | - Conv(16-3x3-2) | /LeakyReLU |
| | | - Conv(16-3x3-2) | /LeakyReLU |
| | MLP Architecture for $e_t(i,j)$ | FC(1296,256) | /ReLU |
| | | - FC(256,128) | /ReLU |
| | | - FC(128,$|\mathcal{A}|$) | |
| | MLP Architecture for $\widetilde{\alpha}_t(i,j)$ | FC(1296,64) | /ReLU |
| | | - FC(64,64) | /ReLU |
| | | - FC(64,1) | |
| | $\lambda_{\text{ent}}$ for Loss | 0.001 | |
| A2C | Discount Factor $\gamma$ | 0.99 | |
| | Learning Rate (RMSProp) | 0.0007 | |
| | Number of Parallel Environments | 16 | |
| | Number of Roll-out Steps per Iteration | 5 | |
| | Entropy Regularization of Policy ($\alpha$) | 0.01 | |
| PPO | Discount Factor $\gamma$ | 0.99 | |
| | $\lambda$ for GAE | 0.95 | |
| | Learning rate (Adam) | 0.00001 | |
| | Number of Parallel Environments | 128 | |
| | Rollout Length | 128 | |
| | Number of Minibatches | 4 | |
| | Number of Optimization Epochs | 4 | |
| | Coefficient of Extrinsic and Intrinsic reward | $\beta_1 = 2, \beta_2 = 1$ | |
| | Entropy Regularization of Policy ($\alpha$) | 0.01 | |

Table 5: The list of hyperparameters used for A2C+CoEX in each game. For the four games where there is no change of high-level visual context (FREEWAY, FROSTBITE, QBERT and SEAQUEST), we do not include $c$ in the state representation $\psi(s)$, hence there is no $\tau$. The same values of $\tau$ are used in PPO+CoEX.

| Games | $\beta_1$ in A2C+CoEX | $\beta_2$ in A2C+CoEX | $\beta_1$ in A2C | $\tau$ for clustering |
|---|---|---|---|---|
| FREEWAY | 10 | 10 | 10 | - |
| FROSTBITE | 10 | 10 | 10 | - |
| HERO | 1 | 0.1 | 1 | 0.7 |
| MONTEZUMA'S REVENGE | 10 | 10 | 10 | 0.7 |
| PRIVATEEYE | 10 | 10 | 10 | 0.55 |
| QBERT | 1 | 0.5 | 1 | - |
| SEAQUEST | 1 | 0.5 | 10 | - |
| VENTURE | 10 | 10 | 10 | 0.7 |

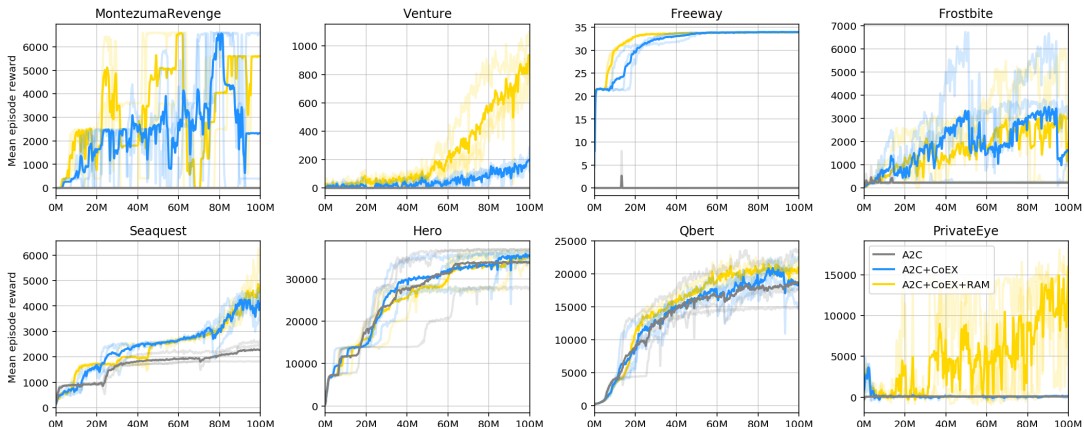

Figure 6: Learning curves on several Atari games: A2C, A2C+CoEX, and A2C+CoEX+RAM.

## C  EXPERIMENT WITH RAM INFORMATION

In order to understand the performance of exploration with perfect representation, we extract the ground-truth location of the agent and the room number from RAM, and then run count-based exploration with the perfect $(x, y, c, R)$. Figure 6 shows the learning curves of the experiments; we could see A2C+CoEX+RAM acts as an upper bound performance of our proposed method.

## D  OBSERVATION EMBEDDING CLUSTERING

We describe the detail of a method to obtain the observation embedding. Given an observation of shape $(84, 84, 3)$, we flatten the observation and project it to an embedding of dimension $64$. We randomly initialize the parameter of the fully-connected layer for projection, and keep the values unchanged during the training to make the embedding stationary.

For the embedding of these observations, we cluster them based on a threshold value $\tau$. The value of $\tau$ for each game with change of rooms is listed in Table 5. If the distance between the current embedding and the center $\mathrm{mean}(c)$ of a cluster $c$ is less than the threshold, we assign this embedding to the cluster with the smallest distance and update its center with the mean value of all embeddings belonging to this cluster. If the distance between the current embedding and the center of any cluster is larger than the threshold, we create a new cluster and this embedding is assigned to this new cluster.

---

**Algorithm 2** Observation Embedding Clustering

---

Initialize parameter $\theta_c$ for context embedding projector if applicable (which is not trainable)
Initialize threshold $\tau$ for clustering
Initialize clusters set $C \leftarrow \emptyset$
**for** each observation $s$ **do**
    ▷ *Get embedding of the observation from the random projection*
    $v \leftarrow f_{\theta_c}(s)$
    ▷ *Find a cluster to which the current embedding fits, if any*
    Find a cluster $c \in C$ with smallest $\|\mathrm{mean}(c) - v\| \leq \tau$, or NIL if there is no such
    **if** $c \neq \mathrm{NIL}$ **then**
        $c \leftarrow c \cup v$
    **else**
        ▷ *if there's no existing cluster that $v$ should be assigned to, create a new one*
        $C \leftarrow C \cup \{v\}$
    **end if**
**end for**

---

In Figure 7, we also show the samples of observation in each cluster. We could see observations from the same room are assigned to the same cluster and different clusters correspond to different rooms.

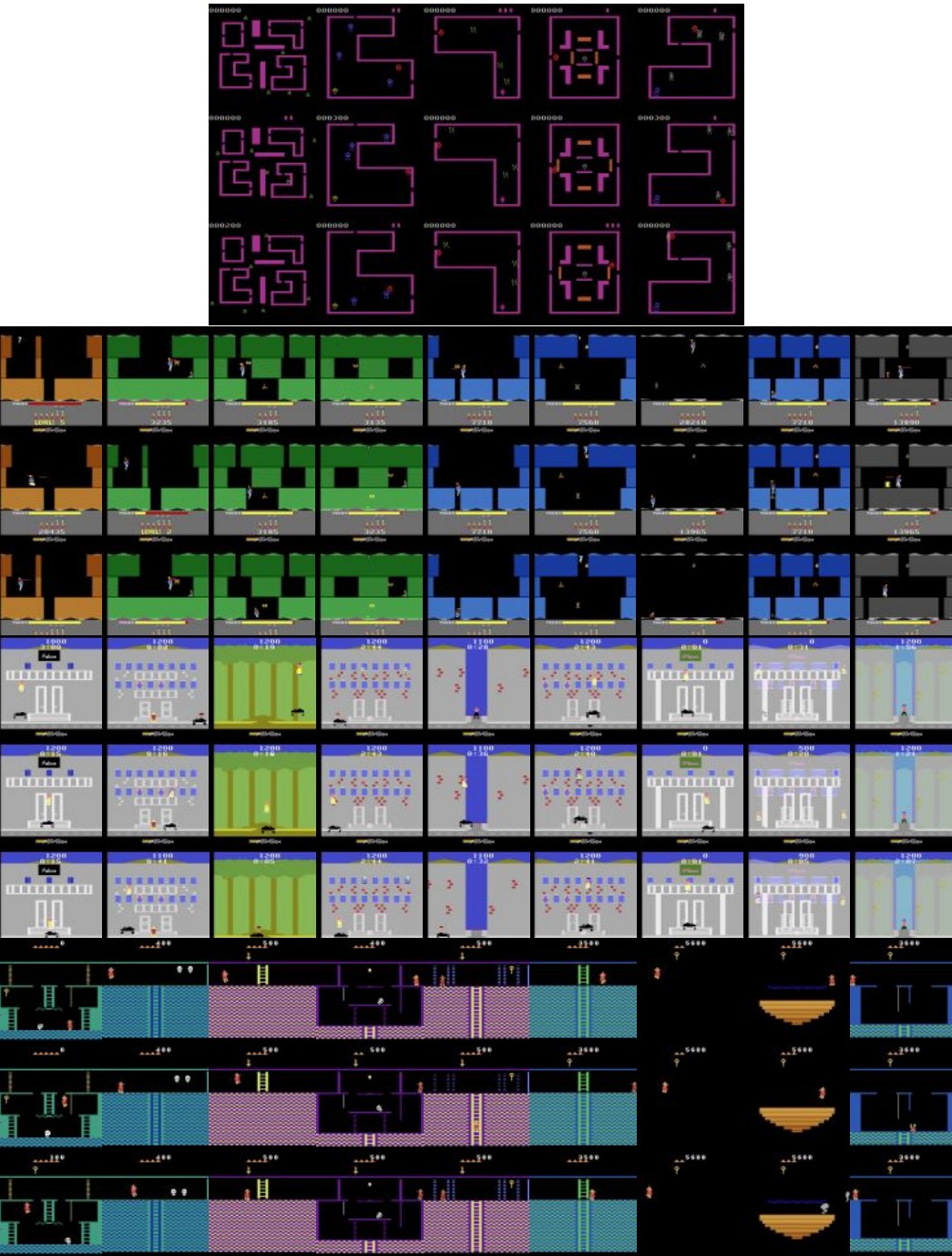

Figure 7: Sample of clustering results for VENTURE, HERO, PRIVATEEYE, and MONTEZUMA'S REVENGE. Each column is one cluster, and we show 3 random samples assigned into this cluster.

# E ABLATION STUDY ON ATTENTIVE DYNAMICS MODEL

We conduct a simple ablation study on the learning objectives of ADM, described in Equation (5). We evaluate the performance of ADM when trained on the same trajectory data under different combinations of loss terms, simulating batches of on-policy transition data to be replayed. The sample trajectory was obtained from an instance of A2C+CoEX+RAM and kept same across all the runs, which allows a fair comparison between different variants. We compare the following four methods:

- ADM (action) : train ADM using $\mathcal{L}_{\text{action}}$ only
- ADM (action, cell) : train ADM using $\mathcal{L}_{\text{action}}$ and $\mathcal{L}_{\text{cell}}$
- ADM (action, ent) : train ADM using $\mathcal{L}_{\text{action}}$ and $\mathcal{L}_{\text{ent}}$
- ADM (action, cell, ent) : train ADM using all losses ($\mathcal{L}_{\text{action}}, \mathcal{L}_{\text{cell}}, \mathcal{L}_{\text{ent}}$)

Figure 8 shows the average distance between the ground-truth location of the agent and the predicted one by ADM during the early stages of training. On MONTEZUMA'S REVENGE, there is only little difference between the variants although the full model worked slightly better on average. On FREEWAY, the effect of loss terms is more clear; in the beginning the agent tends to behave suboptimally by taking mostly single actions only (UP out of three action choices — UP, DOWN, and NO-OP), hence very low entropy $\mathcal{H}(\pi(\cdot|s))$, which can confuse the ADM of telling which part is actually controllable as the action classifier would give correct answer regardless of attention. We can observe additional loss terms help the model quickly correct the attention to localize the controllable object among the uncontrollable clutters with better stability.

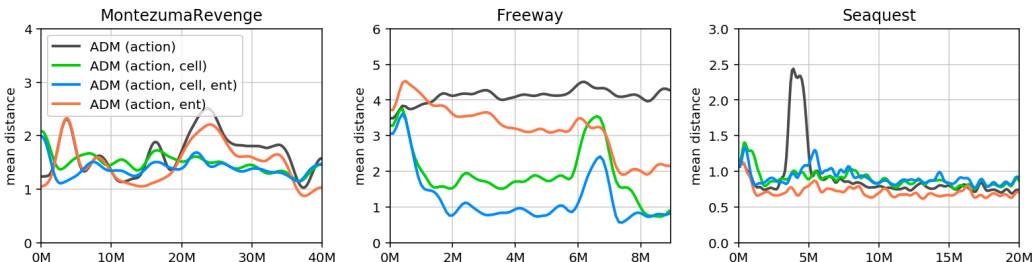

Figure 8: Performance of ADM in terms of mean distance under different loss combinations in early stages, trained using the same online trajectory data. Plots were obtained by averaging runs over 5 random seeds.

In another ablation study, we compare the end performance of the A2C+CoEX agent with the ADM jointly trained under different loss objectives on these three games (MONTEZUMA'S REVENGE, FREEWAY and SEAQUEST). In our experiments, the variant with full ADM worked best on MONTEZUMA'S REVENGE and FREEWAY. The minimal training objective of ADM (i.e., $\mathcal{L}_{\text{action}}$) also solely works reasonably well, but with the combination of other loss terms we can attain a more stable performance.

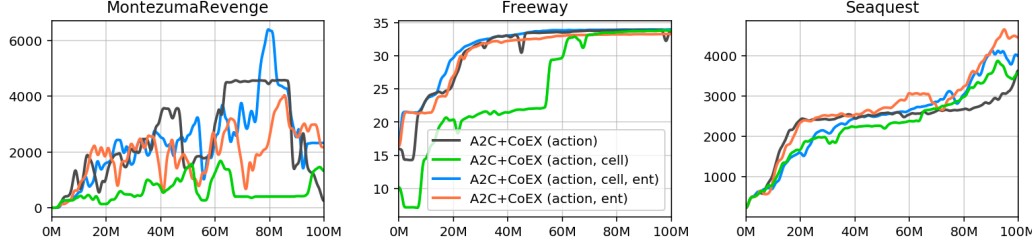

Figure 9: Learning curves of A2C+CoEX with ADM trained under different training objectives. The curve in solid line shows the mean episode over 40 recent episodes, averaged over 3 random seeds.

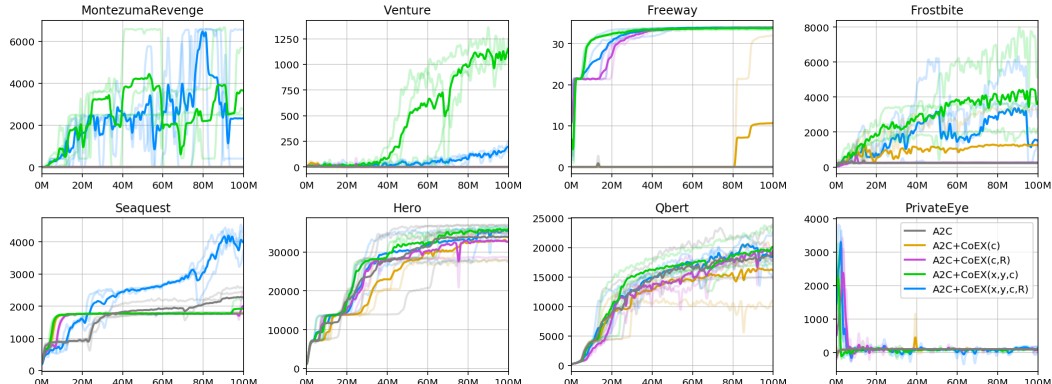

Figure 10: Learning curves for the ablation study of state representation. The exploration algorithm without the contingent region information (purple) performs significantly worse, yielding almost no improvement on hard-exploration games such as MONTEZUMA'S REVENGE, VENTURE, and FROSTBITE. The mean curve is obtained by averaging over 3 random seeds. See Table 6 for numbers.

| Method | Freeway | Frostbite | Hero | Montezuma | PrivateEye | Qbert | Seaquest | Venture |
|---|---|---|---|---|---|---|---|---|
| A2C | 7.2 | 1099 | 34352 | 12.5 | 574 | 19620 | 2401 | 0 |
| A2C+CoEX ($c$) | 10.7 | 1313 | 34269 | 14.7 | 2692 | 20942 | 1810 | 94 |
| A2C+CoEX ($c, R$) | **34.0** | 941 | 34046 | 9.2 | **5458** | 21587 | 2056 | 77 |
| A2C+CoEX ($x, y, c$) | 33.7 | **5066** | **36934** | 6558 | 5377 | 21130 | 1978 | **1374** |
| A2C+CoEX ($x, y, c, R$) | **34.0** | 4260 | 36827 | **6635** | 5316 | **23962** | **5169** | 204 |

Table 6: Summary of the results of the ablation study of the state representation. We report the maximum mean score (averaged over 40 recent episodes) achieved over 100M environment steps, averaged over 3 random seeds.

## F   ABLATION STUDY ON THE STATE REPRESENTATION

We present a result of additional ablation study on the state representation $\psi(s)$ used in count-based exploration. The following baselines are considered:

- A2C+CoEX($c$): Uses only the context embedding for exploration, *i.e.*, $\psi(s) = (c)$.
- A2C+CoEX($c, R$): Uses only the context embedding and the cumulative reward for exploration without contingent region information, *i.e.*, $\psi(s) = (c, R)$.
- A2C+CoEX($x, y, c$): Uses the contingent region information $(x, y)$ as well as the context embedding $c$, however without the cumulative reward component, *i.e.*, $\psi(s) = (x, y, c)$.

One can also consider another baseline similar to A2C+CoEX($c, R$) with $\psi(s) = (x, y, c, R)$, where the location information $(x, y)$ is replaced with random coordinates uniformly sampled from the grid. It ablates the learned contingent regions. However, we found that it performs similarly to the presented A2C+CoEX($c, R$) baseline.

The experimental results are summarized in Table 6 and Figure 10. The variants without contingent regions (*i.e.*, A2C+CoEX($c$) and A2C+CoEX($c, R$) performed significantly worse in most of the games than A2C+CoEX($x, y, c$) and A2C+CoEX($x, y, c, R$) giving little improvement over the A2C baseline. Most notably, in the games with the hardest exploration such as MONTEZUMA'S REVENGE and VENTURE, the performance is hardly better than the vanilla A2C or a random policy, achieving a score as low as zero. The variants with contingent region information worked best and comparable to each other. We observe that using the cumulative reward (total score) for exploration gives a slight improvement on some environments. These results support the effectiveness of the learned contingency-awareness information in count-based exploration.

