# OpenReview forum: "Contingency-Aware Exploration in Reinforcement Learning"
_ICLR.cc/2019/Conference_

### Official Review · AnonReviewer3 · 2018-11-02
**An interesting - but somewhat limited - exploration technique for 2D arcade games**

**Rating:** 7
**Confidence:** 4

**Review:**

This paper investigates the problem of extracting a meaningful state representation to help with exploration in RL, when confronted to a sparse reward task. The core idea consists in identifying controllable (learned) features of the state, which in an Atari game for instance typically corresponds to the position of the player-controlled character / vehicle on the screen. Once this position is known (as x, y coordinates on a custom low-resolution grid), one can use existing count-based exploration mechanisms to encourage the agent to visit new positions (NB: in addition to the x, y coordinates, extra information is also used to disambiguate the states for counting purpose, namely the current score and the state’s cluster index obtained with a basic clustering scheme). To find the position, the algorithm trains one inverse dynamics model per x, y cell on the grid: each model tries to predict the action taken by the agent given two consecutive states, both represented by their feature map (at coordinate x, y) learned by a convolutional network applied to the pixel representation. The outputs of these inverse dynamics models are combined through an attention mechanism to output the final prediction for the action: the intuition is that the attention model will learn to focus on the grid cell with best predictive power (for a given state), which should correspond to where the controllable parts of the state are. Experiments on several Atari games (including Montezuma’s Revenge) indeed show that this mechanism is able to track the true agent’s coordinates (obtained from the RAM state) reasonably well. Using these coordinates for count-based exploration (in A2C) also yields significantly better results compared to vanilla A2C, and beats several previously proposed related techniques for exploration in sparse reward settings.

The topic being investigated here (hard-exploration tasks) is definitely very relevant to current RL research, and the proposed technique introduces some novel ideas to address it, notably the usage of an attention model combined with multiple inverse dynamics models so as to identify controllable features in the environment. The approach seems sound to me and is clearly explained. Combined with pretty good results on well known hard Atari games, I am leaning toward recommending acceptance at ICLR.

I have a few significant concerns though, the first one being that the end result seems quite tailored to the specific Atari games of interest: trying to apply it to other tasks (or even just Atari games with different characteristics) may require significant changes (ex: the assumption that a single region of the screen is being controlled by the agent, the clustering to identify the various “rooms” of a game, and using the total score as a proxy to important state information). I do believe that some components are more general though (in particular the main new ideas in the paper), so this is not necessarily a major issue, but another example of application of these ideas to a different domain could have strengthened the submission.

In addition, even if experiments definitely investigate relevant aspects of the algorithm, I wish there had been an ablation study on the three components of the state representation used for counting (coordinates, cluster and reward). In particular it would be disappointing if similar results could be obtained with just the cluster and reward... even if I do not expect it to be the case, an empirical validation would have been welcome to be 100% sure.

The good results obtained here from exploration alone also beg the question whether this state representation could be useful to train the agent, by plugging it directly as input to the policy network (which by the way may not be trivial due to the co-training, but you get the idea). I realize that the focus of the paper is on exploration, and this is fine, but it seems to me a bit of a waste to build such a powerful state abstraction mechanism and not give the agent access to it. I was surprised that it was not at least mentioned in the discussion or conclusion. Note by the way that the conclusion says the agent “benefits from a compact, informative representation of the world”, which can be misinterpreted as using it in its policy.

Regarding the algorithm itself, one potential limitation is the fact that the inverse dynamics models rely on a single time step to identify the action that was taken. This means that they can only identify controllable state features that change immediately after taking a given action. But if an action has “cascading” effects (the immediate state change causing further changes down the road), there may be other important state features that could be controlled (across longer timesteps), but the algorithm will ignore them (also, in a POMDP one may need to wait for more than one timestep to even observe a single change in the state). I suspect that a more generic variant of this idea, better accounting for long term effects of actions, may thus be needed in order to work optimally in more varied settings.

Finally, I believe more papers deserve to be cited in the “Related Work” section. In particular, the idea of controlling features of the environment, (even if not specifically for exploration), has also been explored in (at least) the following papers:
- “Reinforcement Learning with Unsupervised Auxiliary tasks” (Jaderberg et al, 2017)
- “Feature Control as Intrinsic Motivation for Hierarchical Reinforcement Learning” (Dilokthanakul et al, 2017)
- “Independently Controllable Factors” (Thomas et al, 2017)
- “Disentangling Controllable and Uncontrollable Factors of Variation by Interacting with the World” (Sawada, 2018)
Relying on the position of the agent on the screen to drive exploration in Atari games has also been used in: “Deep Curiosity Search: Intra-Life Exploration Improves Performance on Challenging Deep Reinforcement Learning Problems” (Stanton & Clune, 2018)

Other remarks:
- Please share the code if possible
- In the Introduction, the sentence “it is still an open question on how to construct an optimal representation for exploration” seems to repeat “there is an ongoing open question about the most effective way of using neural network representations for exploration” => I wonder if one was supposed to replace the other?
- On p.2, last line containing citations: Pathak et al should be in the parentheses
- Please explicitly refer to Fig. 1 (Right) in 3.1
- On p.4, three lines above eq. 5, there is a hat{alpha} that should probably be hat{a}
- Is the left hand side L in eq. 5 the same as L^inv in Alg. 1? If so please use the same notations
- “privious” work in 3.2
- In 3.2 please briefly explain what psi is going to be. It is a bit confusing to have it appear “out of nowhere“, with no details on how it is constructed.
- Please explain what the different shades mean in Fig. 2-3
- In Table 2’s caption please add a reference for DQN-PixelCNN. Also what do the star and cross symbols mean next to the algorithms’ names?
- “coule” at end of 4.6
- The “Watson” citation is duplicated in references
- Why are there games with no tau in Table 4? Is it because there was no such clustering on these games? (if yes, that was not clear in the paper). And how was tau chosen for other games? (in particular I want to make sure the RAM state was not used to optimize it)

Update 2018-11-23: I am reducing my rating to 5 (from 6) due to the absence of author response regarding a potential revision addressing my comments/questions as well as those from other reviewers

Update 2018-11-27: I am increasing my rating to 7 (from 5) after the authors responded to reviewers' comments and uploaded a revised version of the paper

---

> ### Author Response · Authors · 2018-11-27
> **Response to Reviewer 3 (Part 1/2)**
>
>
> Dear Reviewer 3,
>
> We appreciate your positive, constructive, and detailed feedback. Our impression was that the rebuttal deadline is extended until November 26 per emails sent from the PCs. We apologize for not submitting the response earlier, as we have been using the extra time from the three-day extension of the revision period to prepare the best version of our response. Below we answer questions and address the concerns mentioned in the review. Please take a look at the revised draft for minor corrections and more related work. Please let us know if this addresses your points; we are happy to provide additional responses/information upon request.
>
> [Specificity of domain]
> Our experiments focus on 2D Atari games as they are popular in the RL community; however, the proposed high-level ideas are more general. We also briefly describe how our method can be extended to address your points.
>
>  > Regarding applicability to different (e.g., non-Atari) environments: The idea of contingency awareness is applicable to continuous control problems as well, e.g., environments with continuous actions and image observations (e.g. rendering of 3D physics-based fully-observable environments from camera, such as AntMaze [Frans et al., ICLR 2018 / Nachum et al., NeurIPS 2018]). In such domains we can still discover controllable aspects out of observations via an attention mechanism by exploiting the correlation between actions and pixels, and then apply a similar exploration technique for the agent.
>
>  > Regarding the assumption that a single region of the screen is being controlled by the agent: To deal with multiple controllable entities in the environment, one can extend our ADM with multiple attention heads, which could identify and track multiple controllable entities. In this case we could enrich the state representation for exploration to include information about multiple objects.
>
>  > Regarding the clustering assumption: we used clustering to identify the context information (e.g., “rooms”), but one can alternatively use different methods to obtain such information, e.g., autoencoder-based distributed representation, and concatenate with the contingent-region information for improving exploration in sparse-reward problems.
>
>  > Regarding using the total score as a proxy to important state information: In environments with sparse rewards it may be natural to assume that collecting a non-zero reward may indicate an important change of context or environmental information (e.g., obtaining a key in Montezuma’s Revenge). The addition of total score as extra state information improved the performance for Montezuma’s revenge. However, for other games, our method was still able to achieve high performance without such total score information. (Please see our ablative studies for details.) We will deemphasize the importance of this component in the final version. Thank you for your insightful comments.

---

> > ### Author Response · Authors · 2018-11-27
> > **Response to Reviewer 3 (Part 2/2)**
> >
> >
> > [Ablative Studies]
> > We conducted an ablation study on the state representation by exploring variants of A2C+CoEX without the predicted location information. We have added it in the Appendix F. To briefly summarize the result: as expected the variants without contingency-region information (especially the (c,R) baseline) perform much worse than the one with contingent region information. It is common for these variants to achieve almost no reward on Montezuma’s Revenge and Venture, where the reward is extremely sparse. This demonstrates that the contingent region information indeed plays an important role in count-based exploration.
> >
> > Method                       | Freeway | Frostbite | Hero | Montezuma | PrivateEye | Qbert  | Seaquest  | Venture
> > A2C                             | 7.2        | 1099       | 34352 | 12.5           | 574            | 19620 | 2401         | 0
> > A2C+CoEX (c)             | 10.7      | 1313       | 34269 | 14.7           | 2692          | 20942 | 1810         | 94
> > A2C+CoEX (c; R)        | 34.0       | 941         | 34046 | 9.2            | 5458          | 21587  | 2056        | 77
> > A2C+CoEX (x; y; c)     | 33.7       | 5066      | 36934 | 6558          | 5377          | 21130 | 1978         | 1429
> > A2C+CoEX (x; y; c; R) | 34.0       | 4260      | 36827 | 6635          | 5316          |23962  | 5169         | 204
> > Table 5: Summary of results for the ablation study: the maximum mean scores (averaged over 40 recent episodes) achieved over 100M environment steps of training.
> >
> >
> >
> > [Providing the policy with learned representation]
> > We first note that one can obtain a better function approximation by using this representation as an additional input, which is already claimed in (Bellemare et al., 2012). One easy way of providing learned contingency region information is to use it as an additional input to the policy and the value network. In our preliminary experiments this improved the performance only by a small margin, therefore we did not include those results for the clarity of the paper. We believe that taking advantage of contingent regions for policy learning could be more useful in a hierarchical RL setting or in combination with planning, which we plan to explore as a future work.
> >
> >
> > [Long-term prediction of ADM]
> > We agree that one could improve ADM by taking multi-step transitions into consideration as suggested. We can consider extending an inverse-dynamics model to provide a window of state sequences that is a few steps wider and predict the action taken in the middle of the transition (e.g. given x_{t-3:t+2} predict a_t). This might be helpful on more complex environments, but it turns out that 1-step prediction works relatively well for the environments we experimented with. We plan to investigate the extension to multi-step prediction in a future work when dealing with more challenging environments.
> >
> > [Writing & Other Remarks]
> > Thanks for pointing out several typos and other suggestions on writing. We have fixed all of them as well as missing references, related work, etc. Regarding Table 2, there were unnecessary star and cross symbols used for denoting different steps which are now removed.
> >
> > [Choice of \tau in clustering]
> > The games with no tau in Table 4 do not have c in the state representation because there is no change of high-level visual context (objects, layouts, etc.) in these games. We did not use RAM to tune the hyperparameter but chose a reasonable value of \tau in the range [0.5, 0.8] based on visual inspection, such that it would give a sensible clustering result of observation samples collected across different visual contexts. One can tune this hyperparameter more extensively if given enough time/computational resources to find the best \tau to reach the highest score in the game; however tuning of \tau was not our primary concern.

---

> > > ### Comment · AnonReviewer3 · 2018-11-27
> > > **Re: Response to Reviewer 3**
> > >
> > > Sorry, I was not aware of the rebuttal period extension. Thank you for the detailed response and updated revision, I will update my review rating accordingly.
> > >
> > > Regarding the (lack of) generality of the proposed method, I do agree that at high level similar ideas could probably be used in different settings, however this remains hypothetical until actually verified empirically (that's what I meant by "another example of application of these ideas to a different domain could have strengthened the submission").
> > >
> > > As far as the last point is concerned (tau), after quickly browsing through the changes in the new revision I didn't see mentioned in the text that some games were not using the clustering scheme. Please make sure it's clear (it should probably be at least in the caption of Table 4). If I understand correctly this also means that for these 4 games, methods A2C and A2C+CoEX(c) in Table 5 are actually the same and the differences only come from re-running the experiments (in that case maybe using the same numbers, e.g. those from A2C, could avoid some confusion).
> > >
> > > About the new content in the revised version:
> > >
> > > 1) On p.16 (last paragraph), "on these two games" should be "on these three games". You also claim that "full ADM worked best", but that is not the case on Seaquest.
> > >
> > > 2) On p.17, you claim that "the variants without contingent regions (...) [gave] almost no improvement over the A2C baseline", mentioning "Montezuma's Revenge and Venture" as examples: however in Venture both variants (scores 94 & 77) improve on A2C (score 0). It's also interesting to see how removing the reward from psi in Venture helps reach a much better score, do you have any idea why? (maybe it somehow has to do with how scoring works in this game?)
> > >
> > > 3) Please mention the number of seeds in Table 5's caption.

---

> > > > ### Author Response · Authors · 2018-11-30
> > > > **Re: Re: Response to Reviewer 3**
> > > >
> > > > Dear Reviewer 3,
> > > >
> > > > Thank you very much for quickly and carefully going through our response and the updated draft.
> > > >
> > > > Regarding the description of \tau, we will update it in the final version of paper. The caption of Table 4 shall now read: “For the four games where there is no change of high-level visual context (FREEWAY, FROSTBITE, QBERT and SEAQUEST), we do not include c in the state representation ψ(s), hence there is no \tau.”
> > > >
> > > > Regarding Table 5, we note that A2C+CoEX(c) slightly differs from the vanilla A2C even on those games, as it has a decaying exploration bonus at each time step, whereas the vanilla A2C has no bonus reward at all. It can affect the agent’s behavior; for instance, a positive reward at every time step is known to incentivize the agent to survive longer.
> > > >
> > > > Regarding your questions about the new ablation study:
> > > >
> > > > 1) This is our small mistake (sorry, Seaquest was added later), thank you for pointing this out. We have fixed this, which will appear in the final version.
> > > >
> > > > 2) The mean score of 94 and 77 happens as a spike in the early stage of training, but the agent failed to retain the score, yielding almost zero mean reward afterwards (as shown in the plot). We will fix wordings accordingly.
> > > >
> > > > The goal in the game of Venture is basically to navigate the world visiting many different rooms and destroy the enemy, and there is not much benefit going back to a previously explored room (more precisely, after clearing the room: i.e., killing enemies and picking up the score-items). Therefore, exploration with cumulative reward as extra state may not be beneficial.
> > > >
> > > > * More detailed answer: we would like to refer to our previous response on why the cumulative rewards may be useful as extra state information as it can potentially serve as important contextual change (e.g., picking up a key in Montezuma’s Revenge) that may incentivize the agent to revisit previously explored states (e.g., going to the door even if the corresponding state was previously explored without the key). However, in Venture, such revisiting behavior based on the change of cumulative rewards does not yield benefit due to the nature of the game.
> > > >
> > > > 3) We have added the number of seeds in Figure 9 and Table 5, which will appear in the final version. Thanks for the suggestion!

---

> > > > > ### Comment · AnonReviewer3 · 2018-11-30
> > > > > **Re: Re: Response to Reviewer 3**
> > > > >
> > > > > Thank you for the clarifications

---

### Official Review · AnonReviewer2 · 2018-11-05
**An Important Step Towards Self Awareness for RL Agents**

**Rating:** 7
**Confidence:** 2

**Review:**

This paper introduces contingency-aware exploration by employing attentive dynamics model (ADM). ADM is learned in self supervised manner in an online fashion and only using pure observations as the agents policy is updated. This approach has clear advantages to earlier proposed count based techniques where agent's curiosity is incentivized for exploration. Proposed technique provides an important insight into how to approach such challenging tasks where the rewards are very sparse. Not only it achieves state of the art results with convincing empirical evidence but also authors make a good job of providing details of their specific modelling techniques for training challenges. They make a good job of comparing and contrasting the contingency-awareness by ADM to earlier proposed methods such as intrinsic motivation and self-supervised dynamics model. Overall exposition is clear with well explained results. The proposed idea raises interesting questions for future work.

---

> ### Author Response · Authors · 2018-11-27
> **Response to Reviewer 2**
>
> Dear Reviewer 2,
> Thank you very much for your feedback. We are glad to hear that you find our work insightful and interesting. We have updated the draft to correct small errors and make the exposition of the paper clearer. Please let us know if you have additional comments. We are happy to provide additional information upon request.

---

### Official Review · AnonReviewer1 · 2018-11-09
**Novel idea for exploration in RL, good empirical results, can benefit from more clarity and evidence**

**Rating:** 6
**Confidence:** 3

**Review:**

Summary:

The paper proposes the novel idea of using contingency awareness (i.e. the agent’s understanding of the environment dynamics, its perception that some aspects of the environment are under its control and ability to locate itself within the state space) to aid exploration in sparse-reward reinforcement learning tasks. They obtain great results on hard exploration Atari games and a new SOTA on Montezuma’s Revenge (compared to methods which are also not using any external data). They use an inverse dynamics model with attention, (trained with self-supervision) to predict the agent’s actions between consecutive states. This allows them to approximate the agent’s position in 2D environments, which is then used as part of the state representation to encourage efficient exploration. One of the main strengths of this method is the fact that it achieves good performance on challenging tasks without the expert demonstrations or environment simulators. I also liked the discussion part of the paper and the fact that it emphasizes some of the limitations and avenues for future work.

Pros:
Good empirical results on challenging Atari tasks (including SOTA on Montezuma’s Revenge without extra supervision or information)
Tackles a long-standing problem in RL: efficient exploration in sparse reward environments
Novel idea, which opens up new research directions
Comparison experiments with competitive baselines

Cons:
The choice of extra loss functions is not very well motivated
Some parts of the paper are not very clear

Main Comments:
Motivation of Extra Loss Terms: It is not very clear how each of the losses (eq 5) will help mitigate all the issues mentioned in the paragraph above. I suggest providing more detailed explanations to motivate these choices. In particular, why are you not including an entropy regularization loss for the policy to mitigate the third problem identified? This has been previously shown to aid exploration. I also did not see how the second issue mentioned is mitigated by any of the proposed extra loss terms.
Request for Ablation Studies: It would be useful to gain a better understanding of how important is each of the losses used in equation 5, so I suggest doing some ablation studies.
Cell Loss Confusion: Last paragraph of section 3.1: is there a typo in the formulation of the per cell cross-entropy losses? Is alpha supposed to be the action a? Otherwise, this part is confusing, so please explain the reasoning and what supervision signal you used.
State Representation: Section 3.2 can be improved by adding more details. For example, it is not explained at all what the function psi(s) contains and how it makes use of the estimated agent location. I would suggest moving some of the details in section 4.2 (such as the context representation and what psi contains) earlier in the text (perhaps in section 3.2).


Minor Comments:
Plots: It would be helpful to give more details about the plots. I suggest labeling the axes. Is the x-axis number of frames, steps or episodes? How many runs are used to compute the mean? What do the light and dark colors represent? What smoothing process did you use to obtain these curves if any? Figure 2, why is there such a large drop in performance on Montezuma’s Revenge after 80M? Something similar seems to happen in PrivateEye, but much earlier in training and the agent never recovers.
Tables: I would suggest reporting results in the tables for more than 3 seeds given that these algorithms tend to have rather high variance. Or at least, provide the values for the variance.
Appendix A, Algorithm 1: I believe this can be written more clearly. In particular, it would be good to specify the loss functions that you are optimizing. There seems to be some mismatch between the notation of the losses in the algorithm and the paper. It would also help to define alpha, c, psi etc.
Footnote on page 4: you may consider using a different variable instead of c_t to avoid confusion with c (used to refer to the context representation).
Appendix D, Algorithm 2: is there a reason for which you aren’t assigning the embeddings to the closest cluster instead of any cluster that is within some range?


References:
The related work section on exploration and intrinsic motivation could be improved by adding more references such as:
Gregor et al. 2016, Variational Intrinsic Control
Achiam et al. 2018, Variational Option Discovery Algorithms
Fu et al. 2017, EX2: Exploration with Exemplar Models for Deep Reinforcement Learning
Sukhbaatar et al. 2018, Intrinsic Motivation and Automatic Curricula via Asymmetric Self-Play
Eysenbach et al. 2018, Diversity is all you need: learning skills without a reward function


Final Decision:

This paper presents a novel way for efficiently exploring environments with sparse rewards.
However, the authors use additional loss terms (to obtain these results) that are not very well motivated. I believe the paper can be improved by including some ablation experiments and making some parts of the paper more clear, so I would like to see these additions in next iterations of the paper.

Given the novelty, empirical results, and comparisons with competitive baselines, I am inclined to recommend it for acceptance.

---

> ### Author Response · Authors · 2018-11-27
> **Response to Reviewer 1 (Part 2/2)**
>
>
> (Continued from part 1)
>
>
> [Results]
> We conjecture that the performance drop on Montezuma’s Revenge is mainly due to the instability of the A2C algorithm when it encounters large nonstationary exploration bonus rewards. However in our preliminary experiments, when a stronger and more stable base RL algorithm is used (e.g., PPO), we observe very stable results without such a performance drop. More specifically, using PPO+CoEX on Montezuma’s Revenge we achieve a score >11,000 averaged over 3 runs at 250M environment steps. The performance seems to keep improving as the number of steps increases, whereas the vanilla PPO achieves a score of <100. This suggests that such a high performance is not due to the use of PPO alone. We report the trend (score vs #steps) below:
>
> Test score,  # of environmental steps
> -------------------------------------------
> 5,066 at 100M steps (= 0.4B frames)
> 8,015 at 150M steps (= 0.6B frames)
> 10,108 at 200M steps (= 0.8B frames)
> 11,108 at 250M steps (= 1B frames)
> (Plot) The corresponding learning curve is available at the supplementary web page: http://goo.gl/sNM3ir
>
> To the best of our knowledge this result is above (or equal to) the state-of-the-art performance in Montezuma’s Revenge without using any explicit high-level information such as RAM states (as in SmartHash [Tang et al., NIPS 2018] or any expert demonstrations (e.g. DQfD [Hester et al., 2017]), when compared with work published to date. We will incorporate more comprehensive experiments with PPO and revise the paper for the final version.
>
> In PrivateEye, we observe the instability of performance mainly due to the trick of clipping reward within the range [-1, 1], which is a standard used in DQN and A2C to deal with different scales of environment rewards. Specifically, PrivateEye has a negative raw reward (e.g. -1 at each time step) but the scale of positive and negative rewards are different (i.e., the scale of positive rewards is often much bigger than that of negative rewards). As a result, the agent actually increases the cumulative sum of “clipped” extrinsic rewards (which increases from around -500 to 0, which correspond to raw reward of approximately 3000 and 0 respectively), but the raw episode return drops as shown in Figure 2. Similar behaviors are also observed in (Bellemare et al. 2016).
>
>
> [Appendix (Algorithm 1&2)]
> We have extended the description of loss functions and fixed notation issues as suggested by the reviewer. Regarding the question about the Algorithm 2 (clustering) it also makes sense to assign a frame to the closest cluster [Kulis & Jordan, 2012]. However, based on our experience we observe that there is no significant difference in terms of the agent’s end performance when we use the closest cluster. This is likely because we have chosen \tau so that such a cluster is mostly unique and there would be only very little difference in room assignment. We will update the paper with the results with the algorithm assigning frames to the closest cluster in the final version.

---

> ### Author Response · Authors · 2018-11-27
> **Response to Reviewer 1 (Part 1/2)**
>
>
>
>
> Dear Reviewer 1,
>
> Thank you for the constructive and positive feedback. Please have a look at the revised draft for ablation studies and other improvements. We are happy to provide additional information upon request.
>
>
> [Extra Loss Terms of ADM]
>
> >> Why not include an entropy regularization loss for policy?
> We agree on the importance of entropy regularization for policy optimization. In fact, in our submission, the standard entropy regularization term H(pi(a|s)) was already included in policy training (we used the default regularization weight 0.01) --- please see Appendix A for details. We have revised the description to make it clearer.
>
> >> How is the second issue (= distribution shift & non i.i.d. training data) mitigated?
> Our goal is to make the ADM model generalize to unseen trajectories. However, if the model is trained only on the trajectories obtained by the current policy, there is a significant risk of overfitting. To prevent this we incorporate different forms of regularization, including attention entropy regularization and policy entropy regularization. We empirically find that this helps the model generalize better. In Appendix E we have included a concrete example on Freeway illustrating the positive impact of additional regularization terms in preventing overfitting.
>
> However, to address this issue more directly, we believe one can potentially incorporate a replay buffer of previous trajectories to optimize the ADM model on off-policy data, or one can train the ADM based on random exploration. We leave this to future work. That being said, we did not observe serious issues with on-line training of the ADM model in our experiments.
>
> >> Ablation Study of ADM.
> We first note that the proposed ADM loss function worked very well on the 8 Atari games considered. That said, there might be other combinations of training objectives that can also work well. Upon your suggestion we have included ablation experiments in Appendix E to study the effect of ADM loss terms. Additional loss terms help to attain better performance and stability of ADM. In environments where the consequence of actions is easily predictable (e.g., Seaquest) the additional regularization may not be necessary. In more difficult games the additional loss terms improve the stability and the generalization of ADM.
>
> [Cell Loss Confusion]
> There was a typo on the cell-wise cross-entropy loss. It was fixed to p(\hat{a} | e) in the revision. Thank you for pointing it out.
>
> [State Representation]
> We have added a small comment on what \psi(s) consists of. We assumed that the construction of \psi(s) can be thought of as an implementation detail in a more general perspective, to simply keep Section 3.2 as concise as possible.
>
> [Plots]
> The x-axis denotes the environment step (100M steps = 400M frames due to the frameskip of 4), and the y-axis denotes the mean reward over recent 40 episodes for each individual run (shown in light curves). The learning curve (shown in dark) is obtained by averaging over 3 random seeds.
>
> (To be continued in part 2)

---

### Public Comment · (anonymous) · 2018-12-11
**Training time for DDQN+ agent is overstated by a factor of 2**

Just a heads-up that you've overstated the training time for Bellemare et al.'s DDQN+ agent by a factor of 2. If you check their Figure 2 and the surrounding text, you'll see that it was only trained for 100m frames, or 25m "environment timesteps" in your terminology.  In Table 2, you've stated that it was trained for 50m environment timesteps. With this in mind, if you compare the first quarter of your Figure 2 to theirs, it seems pretty dubious whether your agent is actually ahead.

Side note: I think it would be better if you quoted training times with the multiplier of 4 throughout, as this is by-and-large the more common time scale used in the literature.

---

> ### Author Response · Authors · 2018-12-13
> **Response**
>
> Thank you very much for your comment. You are correct that the reported performance of DDQN+ is achieved at 25M steps rather than at 50M steps. We will update the table in the final version of the paper. To the best of our knowledge, DDQN+ code is not publicly available and in our experience it was not trivial to replicate the results. On Montezuma’s revenge, very often many methods can reach the score of 2500 quite easily but afterwards they struggle to achieve higher scores (so running the algorithm longer usually doesn’t guarantee further improvement in scores). If the authors can share their code or report their results with more steps on Montezuma’s revenge, we are happy to include it in the table.
>
> Considering that all of the baselines in Table 2 use frameskip of 4, reporting the number of frames (instead of number of steps) does not make a difference in the comparison. However, we will consider reporting the number of frames in the final version.

---

### Meta-Review · Area_Chair1 · 2018-12-13
**Novel approach to exploration with strong empirical validation**

**Confidence:** 4
**Recommendation:** Accept (Poster)

**Metareview:**

The paper addresses the challenging and important problem of exploration in sparse-rewards settings. The authors propose a novel use of contingency awareness, i.e., the agent's understanding of the environment features that are under its direct control, in combination with a count-based approach to exploration. The model is trained using an inverse dynamics model and attention mechanism and is shown to be able to identify the controllable character. The resulting exploration approach achieves strong empirical results compared to alternative count-based exploration techniques. The reviewers note that the novel approach has potential for opening up potential fruitful directions for follow-up research. The obtained strong empirical results are another strong indication of the value of the proposed idea.


The reviewers mention several potential weaknesses. First, while the proposed idea is general, the specific implementation seems targetted specifically towards Atari games. While Atari is a popular benchmark domain, this raises questions as to whether insights can be more generally applied. Second, several questions were raised regarding the motivation for some of the presented modeling choices (e.g., loss terms) as well as their impact on the empirical results. Ablation studies were recommended as a step to resolving these questions Reviewer 3 questioned whether the learned state representation could be directly used as an additional input to the agent, and if it would improve performance. Finally, several related works were suggested that should be included in the discussion of related work.

The authors carefully addressed the issues raised by the reviewers, running additional comparisons and adding to the original empirical insights. Several issues of clarity were resolved in the paper and in the discussion. Reviewer 3 engaged with the authors and confirmed that they are satisfied with the resulting submission. The AC judges that the suggestions of reviewer 1 have been addressed to a satisfactory level. A remaining issue regarding results reporting was raised anonymously towards the end of the review period, and the AC encourages the authors to address this issue in their camera ready version.